# Purple Sweet Potato Powder Containing Anthocyanin Mitigates High-Fat-Diet-Induced Dry Eye Disease

**DOI:** 10.3390/ijms24086983

**Published:** 2023-04-10

**Authors:** Ming-Cheng Chiang, Ying-Chung Liu, Bo-Yi Chen, Dai-Lin Wu, Chia-Lian Wu, Chun-Wen Cheng, Wen-Lung Chang, Huei-Jane Lee

**Affiliations:** 1School of Medicine, Chung Shan Medical University, Taichung 40221, Taiwan; 2Department of Ophthalmology, Cathay General Hospital, Taipei 10687, Taiwan; 3Institute of Medicine, Chung Shan Medical University, Taichung 40221, Taiwan; 4Department of Optometry, Chung Shan Medical University, Taichung 40221, Taiwan; 5Yi-Yeh Biotechnology Co., Taichung 40221, Taiwan; 6Department of Biochemistry, School of Medicine, Chung Shan Medical University, Taichung 40221, Taiwan; 7Department of Clinical Laboratory, Chung Shan Medical University Hospital, Taichung 40201, Taiwan

**Keywords:** purple sweet potato, anthocyanin, high-fat diet, dry eye disease, obesity, oxidative stress, lacrimal gland

## Abstract

Purple sweet potato (PSP) powder with anthocyanins possesses the ability to reduce oxidative stress and inflammation. Studies have presumed a positive correlation between body fat and dry eye disease (DED) in adults. The regulation of oxidative stress and inflammation has been proposed as the mechanism underlying DED. This study developed an animal model of high fat diet (HFD)-induced DED. We added 5% PSP powder to the HFD to evaluate the effects and underlying mechanisms in mitigating HFD-induced DED. A statin drug, atorvastatin, was also added to the diet separately to assess its effect. The HFD altered the structure of lacrimal gland (LG) tissue, reduced LG secretory function, and eliminated the expression of proteins related to DED development, including α-smooth muscle actin and aquaporin-5. Although PSP treatment could not significantly reduce body weight or body fat, it ameliorated the effects of DED by preserving LG secretory function, preventing ocular surface erosion, and preserving LG structure. PSP treatment increased superoxide dismutase levels but reduced hypoxia-inducible factor 1-α levels, indicating that PSP treatment reduced oxidative stress. PSP treatment increased ATP-binding cassette transporter 1 and acetyl-CoA carboxylase 1 levels in LG tissue, signifying that PSP treatment regulated lipid homeostasis maintenance to reduce the effects of DED. In conclusion, PSP treatment ameliorated the effects of HFD-induced DED through the regulation of oxidative stress and lipid homeostasis in the LG.

## 1. Introduction

Dry eye disease (DED) is a multifactorial disorder of the tears and ocular surface that leads to discomfort, visual disturbance, tear film instability, and potential damage to the ocular surface [1]. DED is a common disorder with a prevalence rate ranging from 7% in the United States to 33% in Taiwan and Japan [2]. In addition to aging, autoimmune disease, and hormonal change, metabolic syndrome reduces tear volume [3]. Erdur et al. found that individuals with metabolic syndrome had tear hyperosmolarity and tear film dysfunction [4]. Metabolic syndrome mainly involves body fat accumulation and increased central obesity. Excessive fat accumulation in hypertrophied adipocytes reportedly leads to chronic inflammation, oxidative stress, and dysregulated adipokine secretion [5]. In addition to chronic inflammation, oxidative stress engendered by the accumulation of reactive oxygen species (ROS) is a critical factor for DED development because it induces inflammation of the lacrimal gland (LG), damages the lipid layer of the tear film, and reduces the quality and stability of the tear film [6,7]. Accordingly, therapeutic and nutritional strategies that reduce oxidative stress and inflammation among people with obesity may help prevent DED [8,9,10].

Anthocyanins are a class of polyphenols that are found in many fruits and vegetables, including purple sweet potatoes (PSPs; *Ipomoea batatas* L. cultivar Ayamurasaki). PSP powder containing anthocyanins is used in meal replacement products and as a natural food colorant. An amount of 100 g of the PSP powder contained approximately 8.4 mg of anthocyanin. The clinical potential of anthocyanins as a treatment for DED has attracted considerable research interest. Nakamura et al. demonstrated that anthocyanins can reduce ROS formation in the LG and preserve LG secretory function in a murine model of blink-suppressive DED [11]. A similar outcome was noted in a murine model of benzalkonium chloride-induced DED. Eyes treated with anthocyanins were found to have less corneal erosion compared with those treated with cyclosporin [12]. However, few studies have investigated the association between anthocyanins and obesity-related DED.

To fill this research gap, the present study developed an animal model of high-fat-diet (HFD)-induced DED to investigate the effects and underlying mechanisms of anthocyanins in mitigating DED. We added 5% PSP powder to the HFD (contained 0.42 mg of anthocyanins in 100 g diet) to evaluate the effect of anthocyanins in mitigating DED. Rats were fed a control diet, a HFD, a HFD with PSP powder, or a HFD with a statin drug (atorvastatin) and then compared in terms of changes in LG tissue, tear secretion, and protein levels.

## 2. Results

### 2.1. Effects of Anthocyanin on Body Weight, Visceral Fat, Subcutaneous Fat, and Serum Lipids

This study included 35 four-week-old male Sprague Dawley rats. The rats were acclimated for 1 week (average body weight: 190 ± 10 g). Subsequently, 26 of the rats were fed a HFD. We observed that after 2 weeks, the mean body weight of the rats that were fed the HFD was significantly higher (*p* < 0.05) than that of the rats that were fed the control diet. The rats that were fed the HFD were subsequently randomly allocated to three groups: group H, comprising those that were fed the HFD (*n* = 9); group PSP, comprising those that were fed the HFD with PSP powder (5% of total diet weight; *n* = 9); and group S, comprising those that were fed the HFD with atorvastatin (10 mg/kg of body weight once a day by oral gavage; *n* = 8) (the diet formula shown in Appendix A). An amount of 100 g of the PSP powder contained approximately 8.4 mg of anthocyanin including delphinidin-3-O-glucoside, cyanidin-3-O-glucoside, and petunidin-3-O-glucoside. Atorvastatin can reduce blood lipids and was included in this study to compare its effects with those of PSP. We observed that after 19 weeks, the mean body weight of the rats in group H was significantly higher than that of the rats in the control group (*p* = 0.006). No significant difference in body weight was observed between the rats in group H and those in groups S or PSP, which indicates that atorvastatin and PSP treatment have equally non-significant effects on body weight (Table 1). HFDs increase subcutaneous and visceral fat (defined as perirenal and mesenteric fat). We noted a significant difference between groups H and S in terms of mesenteric adipose tissue. However, group PSP or S did not differ significantly from group H in terms of subcutaneous or other fat (Table 2). In Table 3, the results showed that HFD increased LDL cholesterol, HDL cholesterol, free fatty acid, and glucose, but reduced triglycerides. PSP treatment reduced HDL cholesterol and free fatty acid. Atorvastatin treatment reduced total cholesterol, HDL cholesterol, free fatty acid, and glucose.

### 2.2. PSP Treatment Reduced DED Incidence in HFD Treated Rats

Schirmer’s test is commonly used for DED diagnosis; in this test, the bent end of strip paper is hooked over the center of the lower eyelid to evaluate the capacity of tear secretion by measuring the tear length on the strip paper [13]. In this study, we conducted Schirmer’s test on both eyes of each rat. The mean tear length with outliners excluded is presented in Figure 1. The rats in group H had significantly less tear secretion than did those in the control group (Kruskal–Wallis test, *p* < 0.01). The rats in group PSP, but not those in group S, had a significantly more tear secretion (Kruskal–Wallis test, *p* < 0.01) than did those in group H. These results indicate that among the rats on the HFD, PSP treatment preserved tear secretion capacity.

Corneal lissamine green staining (1%, *w*/*w*) is a useful tool for detecting epithelial erosions on the ocular surface. DED is commonly identified through punctate epithelial erosion staining with green color [14]. In this study, lissamine green staining was performed on both eye of each rat in each group before they were sacrificed (Figure 2A). The number of eyes with positive signals was as follows: one in the control group (*n* = 18), eight in group H (*n* = 18), six in group S (*n* = 16), and two in group PSP (*n* = 18) (Figure 2B). The odds ratio for epithelial erosion in group H relative to the control group was 13.6 (chi-square test, *p* = 0.007). PSP treatment showed a significant reduction in epithelial erosion among the rats on the HFD (chi-square test, *p* = 0.025).

### 2.3. PSP Treatment Improved LG Function in HFD-Treated Rats

LG atrophy is defined as a diminished volume of acinar cells under certain pathological conditions [15]. In Figure 3A, LG atrophy occurred in HFD-treated rats while reduced in PSP group. The quantification results showed that the control group and group H differed significantly in terms of atrophy area (Kruskal–Wallis test, *p* = 0.015), demonstrating a larger LG atrophy area in HFD-treated rats. However, the results showed that atorvastatin was not effective in ameliorating HFD-induced LG injuries (Kruskal–Wallis test, *p* = 0.198). We noted a significant difference between groups H and PSP (Kruskal–Wallis test, *p* = 0.04), indicating that PSP treatment decreased LG atrophy. The pixel densities in the mean areas of LG atrophy were shown in Appendix A.

The LG is an exocrine tubuloacinar gland that secretes the aqueous layer of the tear film. The LG is surrounded by the LG epithelium, which is composed of ductal, acinar, and myoepithelial cells (MECs). MECs express α-smooth muscle actin (α-SMA) and can contract to expel the secretions of exocrine glands. Furthermore, MECs play a crucial role in maintaining glandular structural integrity and transporting metabolites to secretory cells [16]. We performed immunohistochemistry staining to detect the expression of α-SMA in order to quantify the influence of the HFD on MECs. As shown in Figure 4, α-SMA decreased in the group of HFD treatment, while PSP treatment restored α-SMA level obviously. The quantification results clearly illustrated the distribution of α-SMA-positive area in each group. (Figure 4B). Relative to that in the control group, α-SMA expression was significantly lower in group H (Kruskal–Wallis test, *p* < 0.001). Moreover, α-SMA expression differed significantly between groups PSP and H (Kruskal–Wallis test, *p* < 0.001). However, α-SMA expression did not differ significantly between groups S and H (Kruskal–Wallis test, *p* = 0.13).

Aquaporin-5 (AQP5) is a water channel protein encoded by AQP5, playing a role in the generation of saliva, tears, and pulmonary secretions. To evaluate LG function, AQP5 expression was examined. Relative to that in group H, AQP5 expression was preserved in group PSP. Moreover, AQP5 expression did not differ significantly between groups H and S (Figure 5).

### 2.4. PSP Treatment Reduced Oxidative Stress in the LGs of HFD-Ded Rats

Superoxide dismutase (SOD) are a group of metalloenzymes capable of catalyzing the dismutation of superoxide anion free radicals into molecular oxygen and hydrogen peroxide (H_2_O_2_) and reducing the levels of free radical. In this study, we randomly selected three LG specimens from each group for assessment. The expression of SOD-1 in each specimen is displayed in Figure 6. The mean SOD-1 expression level was weakest in group H. Relative to that in group H, SOD-1 expression was elevated in group PSP.

Hypoxia-inducible factor-1 alpha (HIF-1α) is a master regulator of cell response to hypoxia and oxidative stress [17]. A previous study indicated that HIF-1α inhibition ameliorated obesity and insulin resistance [18]. We observed that relative to that in group H, HIF-1α expression was lowered in group PSP but not in group S (Figure 6).

Inhibitor of nuclear factor kappa-B (IκB) is a key regulatory factor that inhibits nuclear factor kappa-light-chain enhancer of activated B cell (NF-κB)-related inflammatory pathways [19]. This study revealed that relative to that in the control group, the IκB level was lower in group H. Moreover, relative to those in groups H and S, the IκB level was higher in group PSP. This indicates that PSP treatment reduced HFD-induced oxidative stress and had anti-inflammatory effects in HFD-treated rats.

### 2.5. PSP Treatment Increases the Level of ATP-Binding Cassette Transporter A1 and Acetyl-CoA Carboxylase 1

ATP-binding cassette transporter 1 (ABCA1) is a transporter protein with two characteristic ATP-binding domains and 12 transmembrane domains that form a channel-like structure for transport; therefore, it constitutes a crucial regulator of cellular cholesterol and phospholipid homeostasis [20,21]. In this study, we quantified the levels of ABCA1. Our results revealed that ABCA1 levels were highest in group PSP (Figure 7).

Acetyl-CoA carboxylase 1 (ACC1), a rate-limiting enzyme in the biogenesis of long-chain fatty acids, is a complex multifunctional enzyme system that catalyzes the carboxylation of acetyl-CoA to malonyl-CoA [22]. Similar to the findings regarding the levels of ABCA1, the levels of ACC1 were notable only in group PSP and were extremely low in the other groups (Figure 7).

## 3. Discussion

DED is a common disorder worldwide, and various factors are associated with its development and progression. This study is the first to investigate the ameliorative effects of PSP treatment on the ocular surface changes caused by DED in HFD-fed rats. Previous studies have shown an association between DED and a HFD [8,10], suggesting that HFD-induced oxidative stress may be a key contributing factor for DED. Nevertheless, clarifying the role of PSP treatment in DED at the functional and microscopic levels is imperative.

The findings of the present study indicate that PSP treatment could reduce oxidative stress accumulation, regulate the activity of inflammatory pathways, and ameliorate HFD-induced DED at both histological and biomolecular levels. We used Schirmer’s test to evaluate LG function. Our results reveal that tear secretion capacity was significantly higher in group PSP than in group H. However, tear secretion capacity in the control group was the same as that in group PSP (*p* = 0.862), signifying that PSP supplementation preserves tear secretion capacity in rats with HFD-induced DED. By contrast, atorvastatin had no meaningfully beneficial effect and did not preserve tear secretion capacity. In addition, we conducted corneal lissamine green staining to detect ocular surface lesions, which are a symptom of DED. According to the results, the HFD was related to the presence of DED; the odds ratio for group H relative to the control group was 13.6. However, group PSP had a significantly lower incidence of ocular surface lesions than did group H, demonstrating that PSP treatment attenuates HFD-related ocular surface injury. We also considered LG atrophy as a representative pathological change in our study. A previous study applied transmission electron microscopy and found that the degree of LG atrophy in patients with Steven–Johnson syndrome-related DED varied [23]. Similarly, our study revealed that the mean area of LG atrophy was largest in group H; the degree of LG atrophy in group PSP was significantly less than that in group H, suggesting that PSP treatment ameliorates HFD-induced oxidative stress in LG tissue.

SMA is the main component of myoepithelial cells and plays a crucial role in maintaining the biophysical function of the LG. A previous study demonstrated that immunohistochemical loss of α-SMA was related to DED [24], and another study noted an obvious decline in LG α-SMA expression in mice after 4 months on a HFD [10]. In the current study, the α-SMA-positive area was significantly larger in group PSP than in group H, indicating that PSP treatment may reduce HFD-related pathological changes in MECs and may serve to ameliorate the effects of DED. Furthermore, α-SMA expression was not significantly different between the control group and group PSP, demonstrating that PSP treatment ameliorates HFD-induced oxidative stress in LG tissue (Kruskal–Wallis test, *p* = 0.909).

We measured AQP5 levels to evaluate changes in water channel protein levels. A previous study indicated that AQP5 may be related to DED; this is because AQP5^–/–^ mice spontaneously exhibited dry eye symptoms and because AQP5 deficiency caused structural changes in LG epithelial cells [25]. Tsubota et al. obtained LG biopsy samples from patients with non-Sjögren DED, patients with Sjögren’s syndrome, and healthy controls. They measured AQP5 levels through ELISA and revealed similar protein expression levels in the three groups, indicating that the pathological change may have been engendered by factors other than protein synthesis. AQP5 should typically be expressed in the apical membrane of acinar cells in the LG through membrane trafficking and targeting. Studies have reported that in patients with Sjögren’s syndrome, AQP5 was rarely labeled at the apical membrane, suggesting a selective defect in AQP5 trafficking in patients with Sjögren’s syndrome [26,27]. However, other research indicated that the mRNA abundance for AQP-5 was significantly lower in acini of rabbits with Sjögren’s syndrome than in that of rabbits without Sjögren’s syndrome [28]. *Dendrobium candidum* extract is a traditional Chinese medicine rich in anthocyanins and can promote the expression of AQP-5 in the exocrine glands of patients with Sjögren’s syndrome [29]. Although our study did not investigate Sjögren’s syndrome, we observed a similar outcome: AQP-5 levels in group PSP were higher than those in group H, implying that morphological alterations in the LG occur following increased oxidative stress.

SOD-1 is responsible for destroying free superoxide radicals in the body [30]. Kojima et al. observed elevated oxidative stress, acinar unit atrophy, monocyte infiltration, decreased secretory function, and apoptotic cell death in the LGs of SOD-1-deficient mice when compared with wild-type mice, demonstrating ROS-mediated LG alterations [31]. Susila et al. reported that DED severity was negatively correlated with blood SOD concentrations [32]. In the present study, we observed that LG SOD-1 levels were higher in group PSP and the control group than in group H, which can be attributed to the accumulation of oxidative stress. SOD-1 expression was highest in group S, possibly due to the obscurity of the last β-actin band, which mistakenly amplified the expression signal.

HIF-1α is a protein that contains a basic helix-loop-helix/PAS domain and serves as a subunit of HIF-1α, which is degraded during normoxia by prolyl hydroxylases. Under hypoxia conditions, HIF-1α degradation is inhibited, consequently resulting in HIF-1α accumulation, which leads to the upregulation of downstream responses including angiogenesis, tissue repair, and cell rejuvenation [33,34]. Wild-type mice with DED exhibited less damage to their secretory structures and minor loss of LG polarities and morphologies when compared with HIF-1α-knockout mice with DED, indicating the depletion of HIF-1α in preventing dry eye–induced acinar cell death in the LG [35]. In this study, we observed that HIF-1α levels were lower in group PSP than in the other groups, indicating that hypoxia was lowest in group PSP. The role of anthocyanins in ameliorating hypoxia-induced inflammation, apoptosis, or cellular injury in SH-SY5Y cells and PC12 cells has been demonstrated in previous studies [36,37]. We believe that the anthocyanins in the PSP powder ameliorated hypoxia-related injury in the LGs of the HFD-treated mice, thereby lowering HIF-1α levels.

NF-κB is a protein complex that regulates innate and adaptive immune functions and is a mediator of inflammatory responses. NF-κB target genes are involved in various cellular responses to stimulation by tumor necrosis factor (TNF)-α, interleukin (IL)-1ß, Toll-like receptors, viruses, or oxidative stress. NF-κB promotes the development of T cells, activation of macrophages, and production of proinflammatory cytokines such as IL-1, IL-6, IL-12, and TNF-α [19,38,39,40]. IκBα is a cellular protein capable of inhibiting NF-κB by masking the nuclear localization signals of NF-κB proteins and preventing them from entering the nucleus. The activation and regulation of NF-κB are tightly controlled by IκBα [41,42]. However, most stimuli can activate NF-κB through IκBα kinase-mediated phosphorylation on its N-terminal serine residues, which then leads to the dissociation of IκBα from NF-κB and degradation of IκBα [43]. Xu et al. studied a mouse model and observed that anthocyanins prevented the phosphorylation of IκBα in a lipopolysaccharide-stimulated J774 cell model in a concentration-dependent manner [44,45]. Similar to the results of the aforementioned study, our findings reveal that the mean IκBα level in group PSP was higher than that in the other HFD-treated groups, indicating that anthocyanins maintain IκBα levels when challenged with HFD-related oxidative stress. In summary, the present study demonstrated that PSP treatment may reduce oxidative stress (as indicated by the preservation of SOD-1 expression in group PSP), ameliorate hypoxia (thereby reducing the accumulation of HIF-1α), inhibit inflammatory responses through the prevention of IκBα phosphorylation, and maintain normal LG structure and function.

ABCA1 promotes cholesterol efflux to extracellular acceptors through which the body removes excess cholesterol from peripheral tissues and delivers them to the liver. Studies have considered intracellular cholesterol accumulation to cause inflammation [46,47,48]. Furthermore, ABCA1 can serve as an anti-inflammatory receptor by activating STAT3, thereby restraining neutrophil production and inhibiting inflammatory responses [49,50]. Studies have already documented the potential capacity of anthocyanins to improve plasma cholesterol levels through ABCA1 upregulation [51,52]. Xia et al. incubated mouse peritoneal macrophages and macrophage-derived foam cells with anthocyanins, cyanidin-3-O-glucoside, and peonidin-3-O-glucoside; they observed that ABCA1 expression was significantly elevated in a dose-dependent manner compared with the control group, and their confocal images revealed that ABCA1 levels increased significantly on the surfaces of cells treated with anthocyanins when compared with untreated cells [53]. Their findings may explain the considerably high levels of ABCA1 in group PSP in our study. PSP treatment upregulates ABCA1 and reduces intracellular lipid accumulation, thereby ameliorating the effects of HFD-induced DED.

Another factor involved in lipid homeostasis maintenance is ACC1, a key enzyme in long-chain fatty acid biogenesis. Previous studies have indicated that supplementation with anthocyanins either decreases ACC1 levels in mRNA or inhibits ACC1 activity [54,55]. In the present study, however, ACC1 levels were highest in group PSP. The tear film is composed of lipid, water, and mucin layers. The lipid layer provides a smooth optical surface for the cornea and reduces tear evaporation from the ocular surface. Lipid layer insufficiency is a leading cause of DED-related excessive evaporation and tear film instability [56,57]. On the ocular surface, the lipid layer is mainly secreted by the meibomian gland. To date, no evidence has indicated that LG would contribute to the lipid layer formation of tear film. In current study, ACC1 detected in LG tissue may not be directedly associated to the tear film stability. Further, increased expression of ACC1 also conflicted with previous studies in which anthocyanins or related-phytochemicals may decrease or inhibit ACC1. For further understanding of this outcome, more research is needed to do to elucidate it.

A meta-analysis of large observational trials revealed that a habitual intake of anthocyanins was associated with protection against cardiovascular disease [58], possibly because anthocyanins improve blood lipid profiles and decrease circulating proinflammatory cytokines. The concentration of anthocyanins in the PSP powder was 8.4 mg per 100 g; the results of the present study indicate that habitual intake of PSPs can prevent DED. In conclusion, a HFD is related to the incidence of DED; specifically, it is related to decreased LG secretory function and an increased incidence of ocular surface lesions. The underlying mechanisms may be HFD-induced oxidative stress and inflammation through the elevation of HIF-1α and the exhaustion of IκBα and SOD-1 due to the hypoxic microenvironment. Furthermore, PSP treatment preserved normal LG structure, significantly reduced the atrophy area, reduced oxidative stress, and inhibited inflammation, as characterized by the stable SOD-1 and IκBα levels and slightly lower HIF-1α levels. PSP treatment also upregulated the level of ABCA1, which decreased intracellular cholesterol and its related oxidative stress. The aforementioned findings are presented graphically in Figure 8.

The present study has several limitations. First, the body weight gains of groups A and S were not significantly higher than that of the control group, but the mean energy intakes of groups H, A, and S were higher than that of the control group. Therefore, the current dry eye model should be HFD-induced rather than obesity-induced. Second, we evaluated the proinflammatory factors such as IL-1β, TNF-α, and NF-κB. Although the results revealed non-significant effects among the groups, other effects such as antioxidation should be examined in the future. Our model could not be used to identify severe DED because it was based on HFD-induced but not chemical injuries or genetic defects. DED is a complex disorder accompanying with a variety of mechanisms such as oxidative stress. PSP powder containing anthocyanins is an antioxidant substance, and our research indeed partially demonstrated its efficacy on a new model of DED. Further research is warranted to determine the role of PSPs or anthocyanin-enriched foods in DED in order to provide therapeutic options for patients with DED.

## 4. Materials and Methods

### 4.1. PSP Powder

PSP powder was a gift obtained from Yi-yeh Biotechnology Co. (Taichung, Taiwan). Briefly, fresh purple sweet potatoes were peeled of the skin then cut into small pieces. Next, the pieces were shadow-dried then mechanically gridded into fine powder, and stored at −20 °C until use. An amount of 100 g of PSP contained 94.1 g carbohydrate, 2.4 g protein, and 0.8 g fat. The anthocyanin content of the PSP powder was determined using high performance liquid chromatography, Dionex Ultimate 3000 series dual low-pressure ternary gradient pump (Dionex Softron GmbH, Germering, Germany) and an Ultimate 3000 series photodiode array detector. Three anthocyanin peaks identified as delphinidin-3-*O*-glucoside, cyanidin-3-*O*-glucoside, and petunidin-3-*O*-glucoside were detected in the chromatogram by diode array detection at 530 nm. After comparison with the HPLC retention times of standard compounds, the anthocyanins were quantified.

### 4.2. Animal Studies

All animal experiments were approved by the Institutional Animal Care and Use Committee at Chung Shan Medical University (Approval no. 2405). Animal care followed the International Guiding Principles for Biomedical Research Involving Animals [60]. Four-week-old male Sprague Dawley rats were obtained from BioLASCO Taiwan (YiLan County, Taiwan) and housed under laboratory conditions (18–23 °C, humidity: 55–60%, 12-h light/dark cycle). After being acclimated for 1 week (average body weight: 190 ± 10 g), the rats were randomly assigned to the following diet groups: control diet (*n* = 9) and HFD (*n* = 26). After 2 weeks, the mean body weight of the rats that were fed the HFD was significantly higher (*p* < 0.05) than that of the rats that were fed the control diet. The rats in the control group were fed a control chow diet that followed the AIN-93M formula, with 10% of energy from fat, 15% from protein, and 75% from carbohydrate. The HFD formula was modified from the previous study [10], with 43% of energy from fat, 12% from protein, and 45% from carbohydrate (shown in Appendix A). The diets were prepared weekly, stored at −20 °C, and thawed to 4 °C every day before being given to the rats for another 19 weeks. All animals were allowed free access to water during this study. Food intake and body weight were recorded daily. At the end of the experiment, the animals were sacrificed through CO_2_ asphyxiation. Visceral (epididymal, perirenal, and mesenteric), and subcutaneous fat tissues, and LGs were collected. One LG was stored in formalin for histological and immunohistological examination, and the other LG was stored at −80 °C until analysis.

### 4.3. Measurement of Teat Volume

Tear volume was measured with Schirmer strips (Alcon Laboratories, Inc., Fort Worth, TX, USA). A 2 mm-wide by 20 mm-long Schirmer strip was cut from Schirmer strips and inserted 1 mm into the lower lid fornix for 2 min. The strip wetting length was measured to the nearest half millimeter. Two eyes in each animal were examined in different days to avoid the interference.

### 4.4. Lissamine Green Stain

Rats were anesthetized with 0.25 mg/kg of Zoteil 50 by intraperitoneal injection. After that, 5 μL of lissamine green (1%, *w*/*w*) was dropped onto ocular surface for 20 s. The lissamine green was removed subsequently, and the eye was washed with normal saline for three times. The ocular surface was taken photo to analyze the lesions.

### 4.5. Histopathology of the LGs

The rats’ LGs were fixed in 10% formalin immediately after removal. For HE staining, 5-μm-thick sections were deparaffinized with xylene, hydrated in a descending series of graded ethanol, stained with hematoxylin for 2 min, rinsed in gentle running water for 2 min, and stained with eosin for 5 s. The slides were examined by a pathologist at 200× magnification. In total, 10 photos were randomly selected from each group, and the atrophy area was quantified by Image J software (ImageJ/Fiji 1.46, obtained from http://imagej.nih.gov/ij/docs/guide, accessed on 5 April 2023) to evaluate the degree of atrophy [61].

### 4.6. Immunohistochemical Analysis

Formalin-fixed and paraffin-embedded eye sections (5 μm) on coated slides were deparaffinized with xylene and rehydrated in a descending series of graded ethanol. Endogenous peroxidase activity was blocked with 0.6% H_2_O_2_. To detect the alteration in smooth muscle in each group, the sections were then incubated with α-SMA antibody (Abcam Co., Cambridge, UK) at a dilution of 1:500, as suggested by the manufacturers, at 37 °C for 1 h, which was followed by detection with the UltraVision Quanto Detection System HRP and DAB Quanto Chromogen and Substrate (Thermo Fishers Scientific, Waltham, MA, USA), counterstaining with Mayer’s hematoxylin, and mounting in glycerin. The sections were observed at 200× magnification. One pathological slide was obtained from each rat and three fields in the slide were randomly selected to take photos under microscopic examination of 200× magnification. Ten photos were selected randomly from each group for quantification by Image J software. The pixel density of positive staining area was taken as the index to quantify and analyze the results.

### 4.7. Western Blotting

LG tissue (0.03 g) was homogenized in 300 μL of phosphate-buffered saline (PBS) containing a protease inhibitor cocktail (Complete, Roche Applied Science, Mannheim, Germany) and a phosphatase inhibitor cocktail (GoalBio, Taipei, Taiwan) with TissuelyserII (QIAGEN, Hilden, Germany) to collect the supernatant. In each group, 50 μg of protein extraction from each specimen were pooled. After quantification, the pooled proteins were analyzed through Western blotting assay. Briefly, the proteins were analyzed through SDS-PAGE and transferred onto a nitrocellulose membrane (Millipore, Bedford, MA, USA), the membrane was blocked with 5% non-fat milk powder in 0.1% Tween-20-Tris-PBS and then incubated overnight with the indicated antibody at 4 °C. Antibodies against SOD-1 (Catalog no. sc-101523, HIF-1α (Catalog no. sc-13515), IκB (Catalog no. sc-1643), ABCA1 (Catalog no. sc-58219), and ACC-1 (Catalog no. sc-271965) were purchased from Santa Cruz (Dallas, TX, USA); AQP5 were purchased from (Catalog no. OSA00089W, Thermo Fishers Scientific, Waltham, MA, MA). They were then incubated with antimouse horseradish peroxidase antibodies (GE Healthcare, Buckinghamshire, UK). The signals were developed through chemiluminescence using the Immobilon Western Chemiluminescent HRP Substrate (Merck Millipore, Danvers, MA, USA) and exposure to ECL hyperfilm in the LAS-3000 luminescent image analyzer (Fujifilm, Tokyo, Japan). Protein quantification was performed through densitometry and using the Multi Gauge (version 2.2; Fujifilm, Stockholm, Sweden).

### 4.8. Statistical Analysis

Values are expressed as the mean ± standard deviation. One-way factorial analysis of variance followed by Duncan’s multiple-range test (Table 1, Table 2 and Table 3) or Kruskal–Wallis test (Figure 3, Figure 4, Figure 5, Figure 6 and Figure 7) was used to analyze intragroup differences, and Student’s *t* test was used to evaluate intergroup differences. *p* < 0.05 was considered statistically significant. Chi-square test was used to analyze the differences of erosion on ocular surface among the groups. All statistical analyses of data were performed using SigmaStat 4.0 or SigmaPlot 10.0 (Sigma Sales Solution Co., Delhi, India).

## Figures and Tables

**Figure 1 ijms-24-06983-f001:**
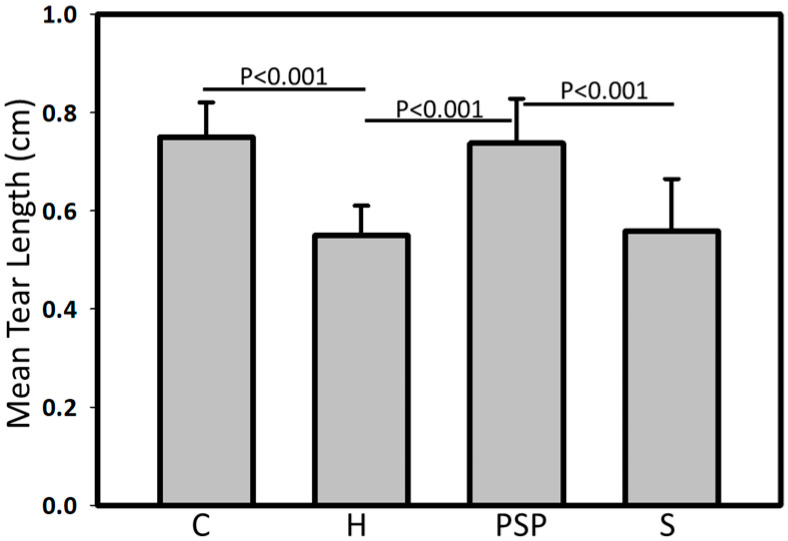
PSP treatment preserved tear secretion capacity in HFD-treated rats. The animals were treated with a high-fat diet (HFD) alone or supplemented with PSP (5%) or atorvastatin (S) for 19 weeks. Before sacrifice, 12 eyes in each group were performed Schirmer’s test to measure the tear secretion. Results were represented as the mean ± SD; *p* values were denoted to show the significant differences.

**Figure 2 ijms-24-06983-f002:**
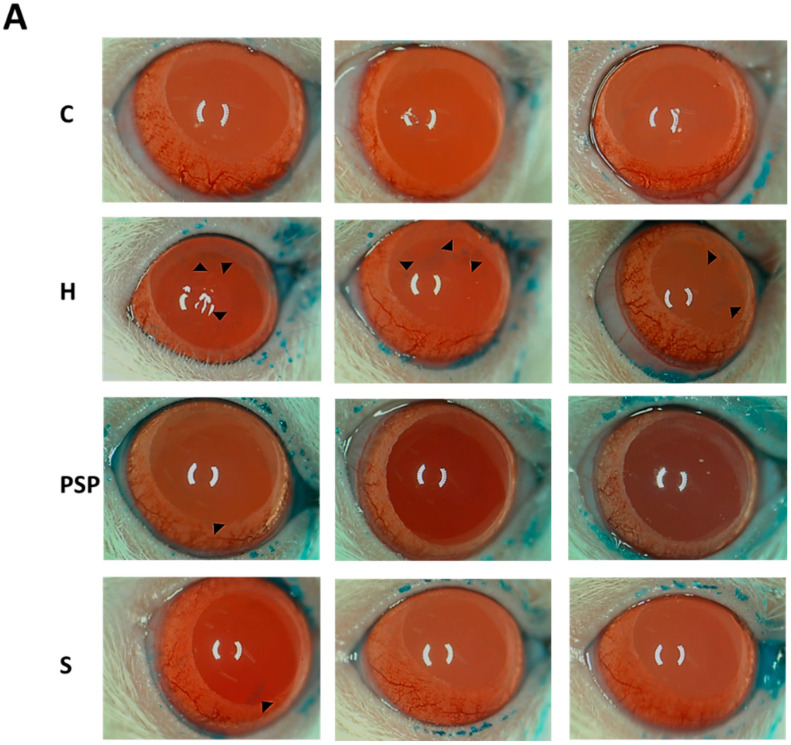
PSP reduced the erosion of ocular surface in HFD-treated rats. The animals were treated with a high-fat diet (HFD) alone or supplemented with PSP (5%) or atorvastatin (S) for 19 weeks. Before sacrifice, both eyes of each rat were stained with 1% (*w*/*w*) of lissamine green. Positive staining area indicated epithelial erosion was labeled with black arrow as shown in (**A**); the counts of positive signal were shown in (**B**). The chi-square test was conducted to analyze the results; *p* values were denoted to show the significant differences.

**Figure 3 ijms-24-06983-f003:**
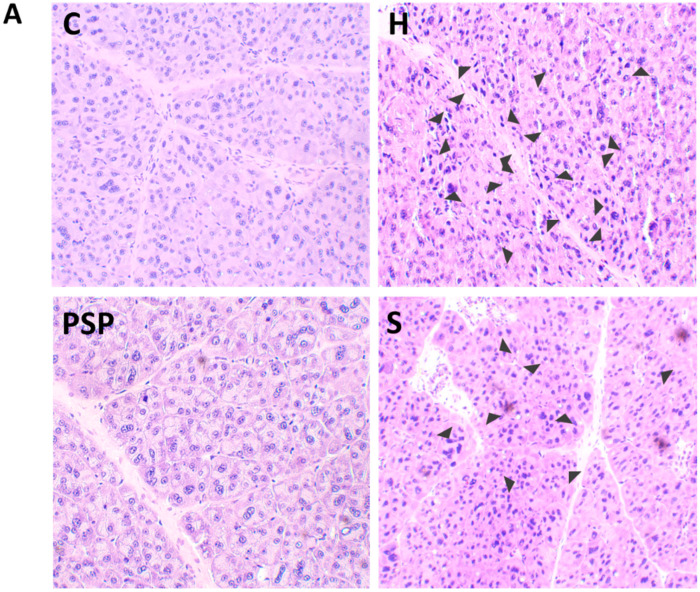
PSP reduced LG atrophy in HFD-*treated* rats. After sacrifice, one eye of each rat was obtained to perform histological examination. (**A**), the atrophy characterized by focal to confluent areas of shrunken and atrophied acini lined by small, low to flattened cuboidal cells as black arrows pointed; (**B**), the results were represented as the mean ± SD to indicate the pixel density of LG atrophy area; *p* values were denoted to show the significant differences.

**Figure 4 ijms-24-06983-f004:**
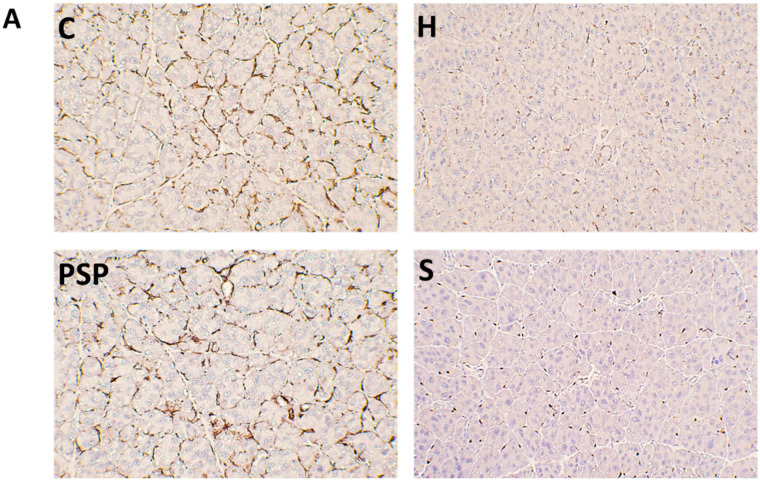
PSP increased α-SMA level in HFD-treated rats. Immunohistological examination was carried out to detect α-SMA level. (**A**), α-SMA-positive area were showed as brown color; (**B**), mean ± SD was represented to indicate the α-SMA-positive area. *p* values were denoted to show the significant differences.

**Figure 5 ijms-24-06983-f005:**
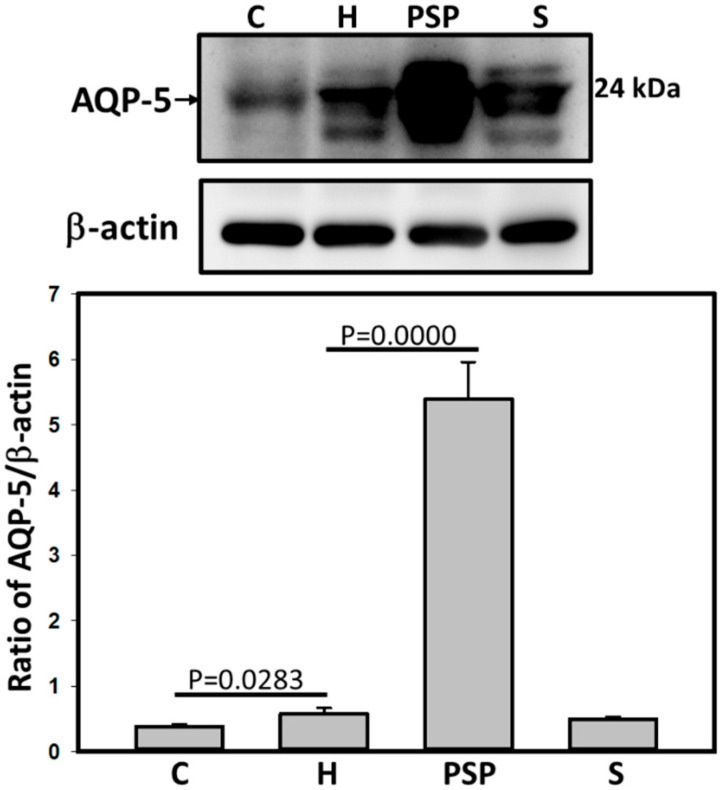
PSP treatment increased AQP5 level. Immunoblot examination was carried out to detect AQP5 level. The β-actin was used as the loading control. The relative image density was quantified by the densitometer. Ratio values calculated from triplicate experiments were represented as the mean ± SD shown in lower panel.

**Figure 6 ijms-24-06983-f006:**
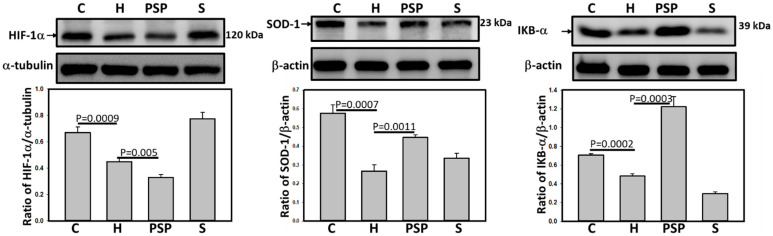
PSP treatment increased SOD-1 and IkB levels but reduced HIF-1α levels. Immunoblot examination was carried out to detect SOD-1, HIF-1α, IkB levels. The β-actin or α-tubulin was used as the loading control. The relative image density was quantified by the densitometer. Ratio values calculated from triplicate experiments were represented as the mean ± SD shown in lower panel.

**Figure 7 ijms-24-06983-f007:**
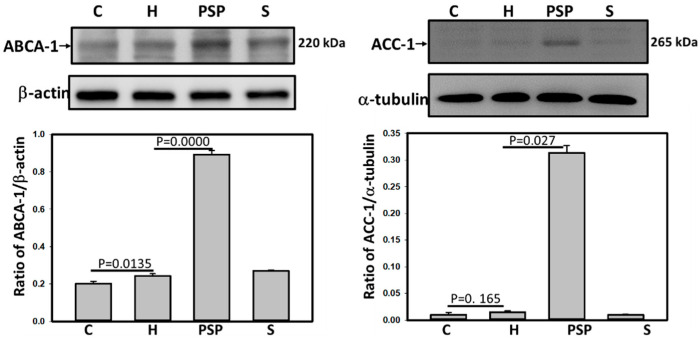
PSP treatment increased ABCA1 and ACC-1 levels. Immunoblot examination was performed to detect ABCA1 and ACC-1 levels. The β-actin or α-tubulin was used as the loading control. The relative image density was quantified by the densitometer. Ratio values calculated from triplicate experiments were represented as the mean ± SD shown in lower panel.

**Figure 8 ijms-24-06983-f008:**
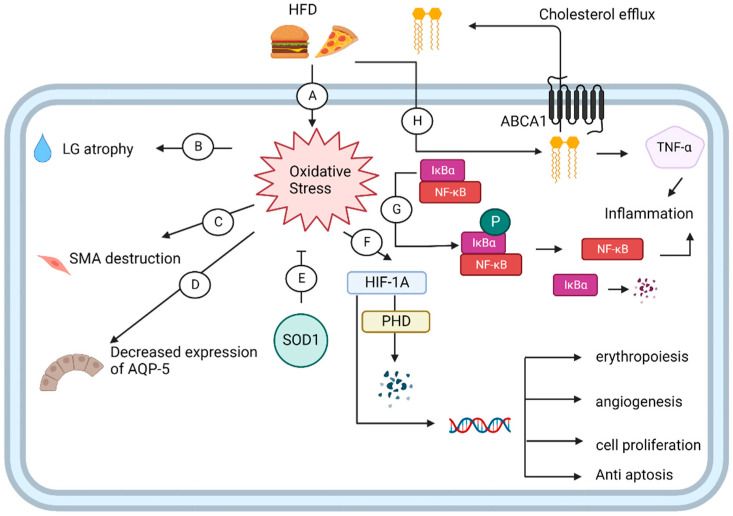
Underlying mechanisms of PSP treatment in ameliorating HFD-induced DED. A, HFD is highly related to accumulation of intracellular oxidative stress, which was a crucial contributing factor toward DED [9,59]. B, Accumulation of oxidative stress leads to LGs atrophy [31], and LGs atrophy area in HFD-fed rats is significantly larger than those of PSP-fed rats. C, Oxidative stress may decrease SMA expression in LGs [60], and α-SMA is lower in HFD-fed rats compared with those with PSP supplement. D, Decrease in AQP-5 mRNA has been found in DED [28]. LGs of HFD-fed rats have lower AQP-5 expression level than that of PSP treatment. E, SOD-1 is preserved more in LGs of PSP-treated rats than that of HFD-fed rats, implying that PSP treatment reduces oxidative stress. F, PSP treatment reduces HIF-1α levels of LGs. G, PSP treatment increases IκBα levels to suppress inflammation. H, PSP treatment increases ABCA1 level.

**Table 1 ijms-24-06983-t001:** Effects of PSP on food intake, energy intake and body weight in Sprague Dawley rats exposed to HFD ^1^.

	Control	HFD	HFD + PSP	HFD + S
Food intake (g/day)	27.7 ± 0.3 ^b^	20.5 ± 0.2 ^a^	20.6 ± 0.2 ^a^	20.6 ± 0.2 ^a^
Energy intake (kcal/day)	79.9 ± 0.9 ^a^	86.1 ± 0.9 ^b^	86.0 ± 0.8 ^b^	86.4 ± 0.8 ^b^
Final body weight (g)	582 ± 38 ^a^	660 ± 59 ^b^	608 ± 35 ^b^	632 ± 39 ^b^

^1^ The animals were treated with a high-fat diet (HFD) alone or supplemented with PSP (5%) or atorvastatin (S) for 19 weeks. The control group was given the control diet. Values (mean ± SD) not sharing a common letter in the same row are significantly different (*p* < 0.05).

**Table 2 ijms-24-06983-t002:** Effect of PSP on body lipid in Sprague Dawley rats exposed to HFD ^1^.

	Control	HFD	HFD + PSP	HFD + S ^2^
Epididymal adipose	19.45 ± 3.4 ^a^	21.8 ± 5.1 ^a^	19.3 ± 5.2 ^a^	24.0 ± 4.7 ^b^
Perirenal adipose	27.6 ± 5.7 ^a^	35.9 ± 8.5 ^b^	34.4 ± 7.2 ^b^	35.9 ± 7.3 ^b^
Mesenteric adipose	9.7 ± 3.4 ^a^	12.5 ± 4.1 ^b^	11.9 ± 2.3 ^b^	10.4 ± 3.5 ^a^
Subcutaneous adipose	22.2 ± 4.1 ^a^	29.2 ± 11.1 ^b^	29.8 ± 5.7 ^b^	31.1 ± 5.8 ^b^

^1^ The animals were treated with a high-fat diet (HFD) alone or supplemented with PSP (5%) or atorvastatin (S) for 19 weeks. The control group was given the control diet. Values (means ± SD, *n* = 8–9) not sharing a common letter in the same row are significantly different (*p* < 0.05). ^2^ mg/g body weight.

**Table 3 ijms-24-06983-t003:** Effects of PSP on the blood biochemistry values in Sprague Dawley rats exposed to HFD ^1^.

	Control	HFD	HFD + PSP	HFD + S ^2^
Triglycerides (mg/dL)	137.0 ± 64 ^a^	70 ± 30 ^bc^	79.0 ± 15 ^bc^	93 ± 31 ^b^
Total cholesterol (mg/dL)	63.0 ± 10.4 ^a^	64.8 ± 17.5 ^a^	59.9 ± 4.3 ^b^	58.3 ± 12.8 ^b^
LDL cholesterol (mg/dL)	5.11 ± 1.5 ^a^	6.71 ± 1.4 ^b^	6.89 ± 0.8 ^b^	6.13 ± 1.6 ^b^
HDL cholesterol (mg/dL)	36.0 ± 5.0 ^a^	40.3 ± 9.9 ^b^	37.5 ± 2.9 ^a^	35.6 ± 8.5 ^a^
Free fatty acid (mg/dL)	0.34 ± 0.05 ^a^	0.40 ± 0.09 ^b^	0.35 ± 0.03 ^a^	0.34 ± 0.05 ^a^
Glucose (mg/dL)	117 ± 17.7 ^a^	136 ± 19.2 ^b^	156 ± 2.9 ^b^	129 ± 24.9 ^a^

^1^ The animals were treated with a high-fat diet (HFD) alone or supplemented with PSP (5%) or atorvastatin (S) for 19 weeks. The control group was given the control diet. ^2^ Values (means ± SD, *n* = 8–9) not sharing a common letter in the same row are significantly different (*p* < 0.05).

## Data Availability

Not applicable.

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
