# Peer review of "Purple Sweet Potato Powder Containing Anthocyanin Mitigates High-Fat-Diet-Induced Dry Eye Disease"

_ijms, 2023, doi:10.3390/ijms24086983_

Round 1
Reviewer 1 Report (New Reviewer)
Overall, the data recollected from this study is interesting. There is some information in this manuscript that requires further support as listed below.
1. The authors should explain the justification for using 5% of PSP powder that contained 8.4 mg per 100 g of anthocyanins.
2. Can the authors explain the rationale for using just male Sprague Dawley rats.? Is Dry Eye Disease predominating in the male population?
3. Fig. 6 The author should present a better quality western blot for HIF1 alpha. There is a line after the first sample that looks odd.
Minor:
Please add Erdur et al….in line 43
The authors need to specify the catalog number of each antibody used in their study.
Some of the text in the article is color red- please correct
Author Response
Thank you for your constructive suggestions. The responses has been attached in the document.

Reviewer 2 Report (New Reviewer)
The paper entitled “Purple Sweet Potato Powder Containing Anthocyanin Mitigates High Fat Diet Induced Dry Eye Disease” aims to evaluate the effects pf purple sweet potato anthocyanins against Dry Eye Disease in high fat fed mice. The manuscript is good and interesting, but it has some flaws that need to be corrected before being considered for publication in this journal:
· The use of English needs improvement; the authors should check the entire text during the revision of the manuscript.
· lines 414-418 should be moved to results section.
· lines 427-429 must be deleted, they are the same of the results.
· how did the authors select the dose of purple potato powder?
· how did the authors select the time of supplementation?
· lines 480-481: which is the “…equal volume of protein …”
· the background of WB ABCA-1 is strange, please check.
Author Response
Thank you for your constructive suggestions. The responses has been attached in the document.

Reviewer 3 Report (New Reviewer)
The authors investigated the protective effects of purple Sweet Potato Powder Containing Anthocyanin against High Fat Diet Induced Dry Eye Disease.
My major concern are:
What is the translational relevance of the results: why not administering anthocyanins through eye drops? The authors should clarify their choice of using oral route of administration.
Cornstarch content of SPD is much lower than other diets. Why?
When dietary administration is used, all the experimental dietary groups should be balanced in terms of other nutrients to highlight the possible effects of the test substances. If other nutrients are not adjusted, their relative contribution to the observed effects is not assessable
TG levels of animals treated with a high fat diet were much lower than those of the control group. This result is doubtful. To be checked
Immunoblot of AQP5 shoes a dark band. Is this specific?
Why were results represented as mean ± SD and as box plot for α-SMA?
Line 383-387: please rephrase these sentences since they are not easily comprehensible.
English style and grammar are poor and should be carefully revised. Many typos: eg. Line 99, 103 alone instead of along
Avoid keep repeating the same phrase (e g. abstract, lines 19-21 )
Author Response
Thank you for your constructive suggestions. The responses has been attached in the document.

Reviewer 4 Report (New Reviewer)
I think this manuscript is acceptable for the publication in IJMS.
Author Response
Thank you for the encourage.
Round 2
Reviewer 2 Report (New Reviewer)
The authors have addressed all comments, the paper can be accepted.
Reviewer 3 Report (New Reviewer)
no further comments
This manuscript is a resubmission of an earlier submission. The following is a list of the peer review reports and author responses from that submission.
Round 1
Reviewer 1 Report
The paper contains many red flags.
1: Control diet (data not shown)
2: Images 2A not enough to detect the green staining showing erosions on the ocular surface to prove it;
3: Immunohistochemistry it's really hard to detect what the authors want to show due to the low magnification and comparisons;
4: LG atrophy again, low magnification and not convinced.
5: Western Blotting showing a big issue in quality and results.
Author Response
Thank you for the constructive suggestion. The comments have been responded in the attached file.

Reviewer 2 Report
The manuscript examines the protective effect of purple sweet potato powder (PSP) in a rat model of high fat diet-induced dry eye disease. Lipid-lowering drug atorvastatin was included as a control. Appropriate measures have been performed to assess the protective effects of PSP. Results and discussion address the study objectives well. It is good that the study limitations have also been discussed. However, the presentation of data requires revision. Furthermore, the quality of the Western blots is poor, hence raises the validity of the assay. There is a large variation in the expression of the internal/house-keeping control b-actin between the treatment groups. Several samples in all Western blots shown in Figures 5, 6 and 7 do not even express any b-actin. Is b-actin the appropriate internal control? Bar graphs in Figures 2-7 should have a similar format as that of Figure 1. Data presented in Figures 2-6 do not appear to be mean+/-SD. Present the densitometry data of western blot shown in Figure 7. Define a and b in Tables 1 and 2. State the duration of treatment in the Methods. Was atorvastatin given every day? If so, state the dose as mg/kg/day. Since the rats are given a high-fat diet and atorvastatin is a lipid-lowering drug, was there any assessment on plasma lipid profile in the rats? Figure 4 legend states that three images of a-SMA positive area were captured while the Method specifies that three fields were imaged per section. Amend the legend in Figure 4. States the SD in Line 143-144 and Table 3. The assessment of lipid enzyme may not be sufficient to support the conclusion that PSP interferes with lipid homeostasis in LG. Evaluation of lipoprotein in the tissues with Western blot or immunohistochemistry would strengthen the argument.
Author Response
Thank you for the suggestions and reminders. The comments have been responded in the attached file.

Round 2
Reviewer 2 Report
There are still issues with the expression of the loading control tubulin in Figures 6 and 7. It is unclear if there is sufficient protein loaded into the blot given the poor expression of tubulin. As such, the data shown in both figures do not seem to be valid. It is recommended to present the densitometry data in bar graphs.